# Latent Ranked Bandits

**Subhojyoti Mukherjee** [1]   **Branislav Kveton** [2]   **Anup B. Rao** [3]

## Abstract

We study the problem of learning personalized ranked lists of diverse items for multiple users, from sequential observations of user preferences. The user-item preference matrix is non-negative and low-rank. Existing methods for solving similar problems are based on reconstructing the preference matrix from its noisy observations using matrix factorization techniques, and typically require strong assumptions on the reconstructed matrix. We depart from this standard approach and consider a family of low-rank matrices, where the set of most preferred items of all users is small and can be learned efficiently. Moreover, in contrast to previous approaches, we assume that the preference matrix is non-stochastic, and so our approach is more general. Then we learn to present this set to each user in a personalized manner, in the order of the descending preferences of the user. We propose a computationally efficient algorithm that implements this procedure, which we call latent ranker (LRA). We evaluate the algorithm empirically on several synthetic and real-world datasets. In all experiments, we outperform existing state-of-the-art algorithms.

## 1. Introduction

In this work, we study the problem of learning personalized ranked lists of diverse items for multiple users, from sequential observations of user preferences. We are interested in utilizing latent similarities among users and items to learn these lists much faster than learning a separate ranked list for each user. The key structure in our problem is that the user-item preference matrix is low rank, which is a standard

---

[*]Equal contribution   [1]College of Information & Computer Science, University of Massachusetts Amherst, Amherst, USA [2]Google Research, Mountain View, California, USA [3]Adobe Research, San Jose, California, USA. Correspondence to: Subhojyoti Mukherjee <subho@cs.umass.edu>, Branislav Kveton <bvketon@google.com>, Anup B. Rao <anuprao@adobe.com >.

*Proceedings of the 36th International Conference on Machine Learning*, Long Beach, California, PMLR 97, 2019. Copyright 2019 by the author(s).

assumption in recommender systems (Koren et al., 2009; Ricci, 2011). The learning agent has access to noisy observations of the user-item matrix. It does not have access to either user or item latent factors.

We formalize our learning problem as the following online learning problem. At round $t$, a random user $i_t$ from a pool of $K$ users arrives to the recommender system. The learning agent observes the identity of the user $i_t$, recommends a list of $d$ diverse items $J_t$ from a pool of $L$ items as a response, and observes the preferences of user $i_t$ for *all* recommended items $J_t$. The user-item preference matrix is low-rank at each round $t$, can vary substantially over time, and does not have to be stochastic. The reward of the recommended list is high when highly preferred items of the user are recommended at higher positions. The goal of our learning agent is to compete with the most rewarding diverse list for each user in hindsight.

Our learning model is motivated by a real-world scenario, where the learning agent suggests movies to users and each movie belongs to different movie genres. The agent typically does not observe instantaneous preferences of the user, and therefore suggests multiple movies that may be of interest to the user under different circumstances. A similar model has also been studied in Carbonell & Goldstein (1998) where the goal is to suggest a diversified list to each incoming user that combines relevance to the query as well as novelty. The authors suggest an approach where each item in the list is relevant to the query but also has *"marginal relevance"* or less similarity with previously selected documents and this improves the quality of recommendation.

We make three major contributions. First, we formulate our online learning problem as a latent ranked bandit on low-rank matrices. We identify a family of non-negative low-rank matrices where our problem can be solved statistically efficiently, without estimating the latent factors of the user-item preference matrix. The key structure of our matrix is that the set of optimal items of all users is small and can be learned jointly for all users. Given these items, the problem of learning the optimal order for each user can be solved in the full-information setting and thus is easy. Second, we propose a computationally-efficient algorithm that implements this idea, which we call latent ranker algorithm (LRA). The algorithm has two components, column learning and row ranking, which learn the set of

optimal items of all users and then sort them, respectively. The column learning algorithm is similar to ranked bandits. In particular, we learn the $k$-th most diverse item using a multi-armed bandit, whose rewards are conditioned on the rewards of $k - 1$ previously chosen items. The row learning problem is solved separately for each user. Because it is in the full-information setting, as we observe the individual rewards of all recommended items, we solve it using the weighted majority algorithm. Third, we evaluate `LRA` empirically on several synthetic and real-world problems. Perhaps surprisingly, `LRA` performs well even when our modeling assumptions are violated.

The paper is organized as follows. We introduce necessary background to understand our work in Section 2 and define our online learning problem in Section 3. We propose our algorithm in Section 4 and analyze its regret in Section 5. In Section 6, we evaluate the algorithm empirically. In Section 7, we survey related work. We conclude in Section 8.

## 2. Background

Let $[n] = \{1, \ldots, n\}$ be the set of the first $n$ positive integers. For any two sets $A$ and $B$, we denote by $A^B$ the set of all vectors whose entries take values from $A$ and are indexed by $B$. Let $M$ be any $m \times n$ matrix. We index the rows and columns of matrices by vectors. For any $d$ and $I \in [m]^d$, $M(I, :)$ denotes a $d \times n$ submatrix of $M$ whose $i$-th row is $M(I(i), :)$. Similarly, for any $d$ and $J \in [n]^d$, $M(:, J)$ denotes a $m \times d$ submatrix of $M$ whose $j$-th column is $M(:, J(j))$. Let $\Pi_d$ be the set of all $d$-permutations. For any $\pi \in \Pi_d$ and $d$-dimensional vector $v$, we denote by $\pi(v)$ the permutation of the entries of $v$ according to $\pi$.

We focus on a family of low-rank matrices, which are known as hott topics. We define a *hott-topics matrix* of rank $d$ as $M = UV^\intercal$, where $U$ is a $K \times d$ non-negative matrix and $V$ is a $L \times d$ non-negative matrix that gives rise to the hott-topics structure. In particular, we assume that there exist $d$ rows $J^*$ in $V$ such that each row of $V$ can be expressed as a convex combination of rows $J^*$ and the zero vector,

$$\forall j \in [L] \, \exists \alpha \in A : V(J^*, :)\alpha = V(j, :), \quad (1)$$

where $A = \{a \in [0, 1]^{d \times 1} : \|a\|_1 \le 1\}$.

The matrix $M$ represents preferences of users for items, $M(i, j)$ is the preference of user $i$ for item $j$. The rank $d$ of $M$ is the number of latent topics. The matrix $U$ are latent preferences of $K$ users over $d$ topics, where $U(i, :)$ are the preferences of user $i \in [K]$. Without loss of generality, we assume that $U \in [0, 1]^{L \times d}$. The matrix $V$ are latent preferences of $L$ items in the space of $d$ topics, where $V(j, :)$ are the coordinates of item $j \in [L]$. We assume that the coordinates are points in a simplex, that is $\|V(j, :)\|_1 \le 1$ for all $j \in [L]$. Note that our assumptions imply that

$M(i, j) \ge 0$ for any $i \in [K]$ and $j \in [L]$.

## 3. Setting

We study an online learning to rank problem, which we call a *latent ranked bandit*. At round $t$, the preferences of users are encoded in a $K \times L$ *preference matrix* $M_t = U_t V^\intercal$, where $M$, $U_t$, and $V$ are defined as in Section 2. We assume that user preferences $U_t$ can change with time. A random user $i_t \in [K]$ arrives to the recommender system at time $t$ and we recommend $d$ items $J_t$ to this user. The *reward* for recommending these items is $r_t(i_t, J_t)$, where

$$r_t(i, J) = \max \{\mu(k) \, M_t(i, J(k)) : k \in [d]\} \quad (2)$$

is the reward for recommending items $J$ to user $i$ at time $t$, $J(k)$ is the $k$-th item in $J$, and $\mu(k)$ is the weight of position $k \in [d]$. We assume that higher-ranked positions are more rewarding, $1 \ge \mu(1) \ge \cdots \ge \mu(d) \ge 0$. The learning agent *observes* the individual rewards of all recommended items, $M_t(i_t, J_t(k))$ for all $k \in [d]$.

Since $U_t$ can change arbitrarily over time, the reward in (2) is maximized by lists $J$ with highly rewarding items that are diverse, in the sense that they attain high rewards at different times $t \in [n]$. Because the rewards are weighted by $\mu$, more frequent highly-rewarding items should be placed at higher positions. A remarkable property of our user-item preference matrices $M_t$ is that for any user $i \in [K]$ at any time $t$,

$$\underset{j \in [L]}{\arg \max} \, M_t(i, j) \in J_*,$$

where $J_*$ is defined in (1). Therefore, it is possible to learn all potentially most rewarding items statistically efficiently.

Now we are ready to define our notion of optimality and regret. Let $J_*$ be the hott-topics items in (1) and $\pi_{*,i}$ be their permutation that maximizes the reward of user $i$ in hindsight,

$$\pi_{*,i} = \underset{\pi \in \Pi_d}{\arg \max} \sum_{t=1}^{n} r_t(i, \pi(J_*)).$$

Let $J_t$ be our recommended items at time $t$ and $\pi_{t,i}$ be their permutation for user $i$, both of which are learned. Then our goal is to minimize the expected $n$-step regret,

$$R(n) = \sum_{t=1}^{n} \mathbb{E} \left[ r_t(i_t, \pi_{*,i_t}(J_*)) - r_t(i_t, \pi_{t,i_t}(J_t)) \right], \quad (3)$$

where the expectation is with respect to both randomly arriving users and potential randomness in the learning algorithm.

# 4. Algorithm

We propose *latent ranker algorithm (*LRA*)* for solving the personalized ranking problem. The pseudocode of LRA is in Algorithm 1. LRA has two main components, column learning and row ranking.

The column learning algorithm recommends a list of $d$ columns and is the same as in Radlinski et al. (2008). But we exploit an additional structure in our problem to show that we learn the optimal columns $J_*$. The column learning algorithm are $d$ instances of multi-armed bandit algorithms, which we denote by ColAlg($k$) for algorithm $k \in [d]$. ColAlg(1) learns the most rewarding column on average, ColAlg(2) learns the second most rewarding column on average conditioned on the first learned column, and so on.

The row ranking algorithm permutes columns suggested by the column learning algorithm. It consists of multiple instances of full-information algorithms. More precisely, for each user $i \in [K]$ and set of $d$ columns $J$, we have algorithm RowAlg($i, J$) with $d!$ arms, which correspond to all possible permutations of $J$. The objective of RowAlg($i, J$) is to learn a permutation of $J$ with the highest reward, as measured by (2).

LRA interacts with the environment as follows. At round $t$, a random user $i_t$ is revealed to LRA. Then, in the ascending order of $k \in [d]$, ColAlg($k$) suggests column $\ell_k$. If ColAlg($k$) suggests one of the previously suggested columns $\ell_1, \ldots, \ell_{k-1}$, then $\ell_k$ is chosen uniformly at random from the remaining columns. We denote the vector of $d$ suggested columns by $J_t$. Then RowAlg($i_t, J_t$), the row learning algorithm for user $i_t$ and columns $J_t$, selects permutation $\pi_{t,i_t}$ of $J_t$.

The user is recommended a permuted list $\pi_{t,i_t}(J_t)$ and LRA observes the individual rewards of all recommended items. Then we update both column and row learning algorithms. The reward of the arm in ColAlg($k$), which selects the $k$-th column in $J_t$, is updated as follows. If the arm was not one of the previously suggested columns, its reward is $\max \{M_t(i_t, J_t(a)) : a \in [k]\} - \max \{M_t(i_t, J_t(a)) : a \in [k-1]\}$. Otherwise, we update the initially suggested arm with reward 0. Since LRA observes the individual rewards of all recommended items, we can compute the reward of any permutation of $J_t$ in row $i_t$. These rewards are then used to update RowAlg($i_t, J_t$).

## 4.1. Practical Considerations

The proposed LRA algorithm only has to update/look through $(Kd+d)$ items for each of the $d$ ColAlg and the i-th RowAlg at every timestep $t$. This is in stark contrast to some of the existing matrix completion algorithms which has to reconstruct a $K \times L$ matrix (Sen et al., 2016) or calculate second or third order tensors (Gopalan et al., 2016).

---

**Algorithm 1** Latent Ranker Algorithm (LRA)

1: **Input:** Rank $d$, horizon $n$
2: **for** $k = 1, \ldots, d$ **do** {Initialization}
3:      Initialize ColAlg($k$)
4: **end for**
5: **for all** $i \in [K], J \subset [L]$ such that $|J| = d$ **do**
6:      Initialize RowAlg($i, J$)
7: **end for**
8:
9: **for** $t = 1, \ldots, n$ **do**
10:      User $i_t$ is revealed
11:      **for** $k = 1, \ldots, d$ **do** {Generate response}
12:          $\hat{\ell}_k \leftarrow$ Suggested item by ColAlg($k$)
13:          **if** $\hat{\ell}_k \in \{\ell_1, \ldots, \ell_{k-1}\}$ **then**
14:              $\ell_k \leftarrow$ Random item not in $\{\ell_1, \ldots, \ell_{k-1}\}$
15:          **else**
16:              $\ell_k \leftarrow \hat{\ell}_k$
17:          **end if**
18:      **end for**
19:      $J_t \leftarrow (\ell_1, \ldots, \ell_d)$
20:      $\pi_{t,i_t} \leftarrow$ Suggested permutation by RowAlg($i_t, J_t$)
21:      Recommend $\pi_{t,i_t}(J_t)$
22:      Observe $M_t(i_t, J_t(k))$ for all $k \in [d]$
23:      **for** $k = 1, \ldots, d$ **do** {Update statistics}
24:          **if** $\ell_k = \hat{\ell}_k$ **then**
25:              Update arm $\ell_k$ of ColAlg($k$) with reward

$$\max \{M_t(i_t, J_t(a)) : a \in [k]\} - \\ \max \{M_t(i_t, J_t(a)) : a \in [k-1]\}$$

26:          **else**
27:              Update arm $\hat{\ell}_k$ of ColAlg($k$) with reward 0
28:          **end if**
29:      **end for**
30:      **for all** arms $\pi$ in RowAlg($i_t, J_t$) **do**
31:          Update arm $\pi$ with reward $r_t(i_t, \pi(J_t))$ in (2)
32:      **end for**
33: **end for**

---

Note, that we leave the implementation of the ColAlg and RowAlg to the users. For theoretical guarantees we use non-stochastic algorithm Exp3 as ColAlg and WMA as RowAlg which will be explained in detail in section Section 5. For experimental purposes, stochastic algorithms like UCB1 or thompson sampling (Thompson, 1933), (Thompson, 1935), (Agrawal & Goyal, 2012) can also be used to improve the performance of LRA. This has also been explored in Radlinski et al. (2008) where RBA uses UCB1 for ranking items. Note, that thompson sampling is a Bayesian algorithm that performs better than UCB1 in stochastic setting due to its inherent prior assumptions on the distribution of the feedback.

# 5. Analysis

We sketch the main components of the regret decomposition here and argue that the regret should be bounded. The crucial idea is to decompose the regret of `LRA` into two parts, where `ColAlg` does not suggest $J_*$ and the rest. The first part can be analyzed as follows. `ColAlg` has a sublinear regret, based on a similar analysis to Radlinski et al. (2008). Therefore, our upper bound on the probability that `ColAlg` suggests suboptimal columns, is bounded, and decreases with time horizon $n$.

The second part is analyzed as follows. Conditioned on $J_t = J_*$, the only remaining regret at time $t$ is due to the fact that columns $J_t$ are not ordered optimally for user $i_t$. Since this is a full-information problem where `RowAlg` is the weighted majority algorithm (Littlestone & Warmuth, 1994), the regret due to learning to order $d$ columns for $K$ users is $K$ times that of the weighted majority algorithm with $d!$ arms. Hence, the regret should consists of two main parts. The first part is the regret due to learning the $d$ optimal hott-topics columns with a high probability. The second part, which is due to learning the most rewarding permutations of the $d$ optimal columns for all users.

A key challenge in proving the regret is to handle certain special cases. One particular case might occur in the rank-1 setting, which is a trivial case as every user $i \in [K]$ prefers a single best item $j^* \in [L]$ and there is no ranking of items present. Yet, if the matrix $M(i, j)$ is 0 everywhere except at the $i, j^*$ position for all $i \in [K]$, then regret might scale as $O(\sqrt{KLn})$. Hence, deriving a regret bound for this setting requires more careful analysis and possibly further assumptions on $U$ or $V$.

Another challenge in this setting stems from the fact that rows (or users) are being revealed by nature, and the learner cannot choose them. This makes the regret decomposition and subsequent analysis more difficult. Note, that this setting is in contrast to Katariya et al. (2017), Katariya et al. (2016), and Kveton et al. (2017) settings where the rows are also being selected by the learner.

Note, that if this problem was solved by RBA (Radlinski et al., 2008), the regret would be $O(\sqrt{KLn})$. Similarly, the trivial approach where the optimal columns are learned separately for each user by separate bandit algorithms will also result in a regret of $O(\sqrt{KLn})$.

Finally, we use non-stochastic algorithms for `ColAlg` and `RowAlg` because our environment is non-stationary. In particular, we assume that user preferences $U_t$, and thus rewards, can change over time $t$. In addition, the rewards in `ColAlg(k)` are non-stationary due to chosen columns at higher positions $1, \ldots, k-1$.

# 6. Experiments

In this section, we compare `LRA` to several bandit algorithms in three experiments. The first two experiments are on synthetic dataset where all modeling assumptions hold. The third experiment is on a real-life dataset where we evaluate `LRA` when our modeling assumptions fail. In all our experiments user come uniform randomly over all time $[n]$. All results are averaged over 10 independent random runs.

## 6.1. Evaluated Algorithms

**Independent User Model Algorithms:** In this approach, each user has a separate version of base-bandit algorithm running independent of each other. As base-bandit algorithms we choose two variants of the ranked bandit algorithm (`RBA`) of Radlinski et al. (2008). The two variants of `RBA` uses two types of column learning algorithms, `UCB1` (Auer et al., 2002a) and `Exp3` (Auer et al., 2002b), abbreviated as `RBA − UCB1` and `RBA − EXP3` respectively. `Exp3` is a randomized algorithm suited for the adversarial setting while `UCB1` is the standard algorithm used in the stochastic feedback setting. For `RBA − UCB1`, we choose the confidence interval at round $t$ as $c_{i,j}(t) = \sqrt{\frac{2 \log t}{N_{i,j}(t)}}$ for user $i$ and item $j$. Here, $N_{i,j}(t)$ denotes the number of times the $j$-th item has been observed by the $i$-th user base-bandit algorithm till round $t$. Note, that running independent vanilla `UCB1` and `Exp3` for every user is not feasible. This is because the vanilla versions are guaranteed to find a single best item for each user at rank 1, while `RBA − UCB1` and `RBA − EXP3` will find a diverse list of $d$ best items for each user.

**Matrix Completion Algorithms:** In the matrix completion approach, the algorithms try to reconstruct the user-item preference matrix $M$ from its noisy realization. We implement the widely used non-negative matrix factorization method to reconstruct partially observed noisy matrices. We term the corresponding algorithm as NMF Bandit (`NMF − Ban`). The objective function of `NMF − Ban` is:-

$$\text{minimize} \left\| \hat{M} - \hat{U}\hat{V}^{\mathsf{T}} \right\|_F^2 \text{ with respect to } \hat{U}, \hat{V}$$
$$\text{subject to constraints } \hat{U}, \hat{V} \geq 0$$

where, $\hat{M}$ is the observed noisy matrix of size $K \times L$ which has a low rank $d$, $\hat{U} \in [0, 1]^{K \times d}$ and $\hat{V} \in [0, 1]^{L \times d}$ are estimated non negative matrices which generates $\hat{M}$ such that $\hat{M} \approx \hat{U}\hat{V}^{\mathsf{T}}$. This objective function is minimized by alternating minimization of $\hat{U}$ and $\hat{V}$ till the loss is very low. `NMF − Ban` knows the rank of the matrix $\hat{M}$. This algorithm is explore-exploit in implementation whereby it first explores for $cd(K + L)$ rounds by choosing items for incoming users uniform randomly, where $c$ is an exploration parameter which can be tuned depending on the noise in

the system. We set $c = 10$ in all experiments. Then it reconstructs $\hat{M}$ using the objective function mentioned above. Then over the reconstructed matrix it behaves greedily and suggest $d$ best items based on decreasing order of their preferences for the $i_t$-th user at every timestep $t$.

**Personalized Ranking Algorithms:** In this approach, we evaluate our proposed algorithm latent ranking bandit (LRA) by using two different types of column learning algorithms, Exp3, and UCB1. We term them as LRA − EXP3, and LRA − UCB1 respectively. The row ranking components for all of these algorithms is the weighted majority algorithm (WMA) from Littlestone & Warmuth (1994) which is suited for the full information setting. Note that we only show theoretical guarantees for LRA − EXP3. We initialize the $k$-th column EXP3 with the column exploration parameter $\gamma_k = \sqrt{\frac{L \log L}{n}}$ as stated in Auer et al. (2002b). Similarly, for LRA − UCB1 we use a confidence interval of $c_{k,j}(t) = \sqrt{\frac{2 \log t}{N_{k,j}(t)}}$ for the $k$-th column MAB and $j$-th item. Here, $N_{k,j}(t)$ denotes the number of times the $j$-th item has been observed by the $k$-th column UCB1 algorithm till timestep $t$.

## 6.2. Synthetic Experiment 1

This experiment is conducted to test the performance of LRA over small number of users and items. This simulated testbed consist of 500 users, 50 items, and $\text{rank}(M) = 2$. The vectors spanning $U$ and $V$, generating the user-item preference matrix $M$, are shown Figure 1(a). The users are evenly distributed into a $50 : 50$ split such that $50\%$ of users prefer item 1 and $50\%$ users prefer item 2. The item hott-topics are $V(1, :) = (0, 1)$ and $V(2, :) = (1, 0)$ while remaining $70\%$ of items has feature $V(j', :) = (0.45, 0.55)$ and the rest have $V(j, :) = (0.55, 0.45)$. We create the user feature matrix $U$ similarly having a $50 : 50$ split such that $U(1, :) = (0, 1)$, $U(2, :) = (0.2, 0.8)$ and the remaining $70\%$ users having $U(i, :) = (0, 0.8)$ and $30\%$ users having $U(i', :) = (0.7, 0)$. At every timestep $t$ the resulting matrix $M_t = U D_t V^\intercal$ is generated where $D_t$ is a randomly-generated diagonal matrix. So, $M_t$ is such that algorithms that quickly find the easily identifiable hott-topics perform very well. From Figure 1(b) we can clearly see that LRA − EXP3, and LRA − UCB1 outperforms all the other algorithms. Their regret curve flattens, indicating that they have learned the best items for each user. Independent user model algorithms RBA − UCB1 and RBA − EXP3 perform poorly as the number of items per user is too large and the independent algorithms are not sharing information between them. NMF − Ban performs better than the independent user model algorithms but is outperformed by LRA − EXP3, and LRA − UCB1.

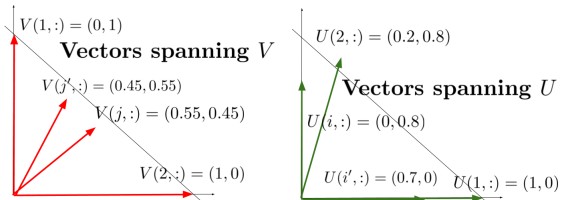

(a) Expt-1: 500 Users, 50 items, Rank 2, User and Item vectors

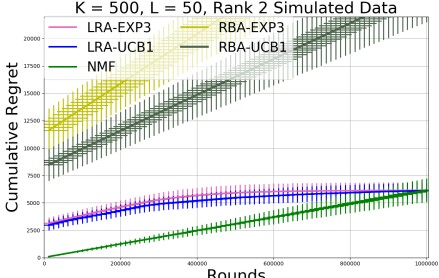

(b) Expt-1: Cumulative regret of different algorithms

*Figure 1.* A comparison of the cumulative regret incurred by the various bandit algorithms.

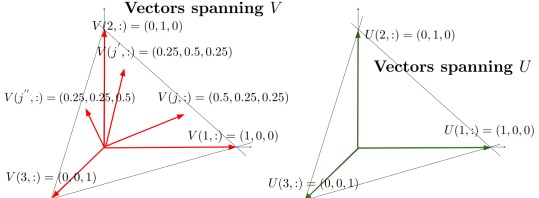

(a) Expt-2: 1500 Users, 100 items, Rank 3, User and Item vectors

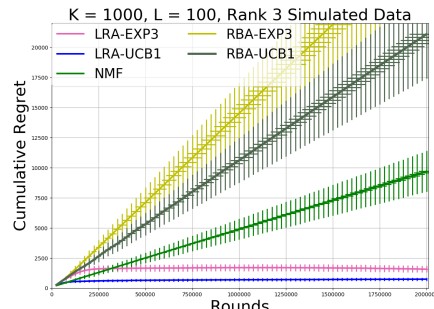

(b) Expt-2: Cumulative regret of different algorithms

*Figure 2.* A comparison of the cumulative regret incurred by the various bandit algorithms.

## 6.3. Synthetic Experiment 2

We conduct the second experiment on a larger simulated database of 1500 users, 100 items and $\text{rank}(M) = 3$. The vectors spanning $U$ and $V$, generating the user-item preference matrix $M$ is shown Figure 2(a). The users are divided into an unequal distribution of $60 : 30 : 10$ split such that

60% of the users prefer item item 1, 30% prefer item 2 and 10% prefer item 3. Hence, in this testbed it is difficult to learn item 3 as it is observed for less number of users. Here, hott-topics are $V(1,:) = (1,0,0)$, $V(2,:) = (0,1,0)$ and $V(3,:) = (0,0,1)$. The remaining 60% of items have feature $V(j,:) = (0.5, 0.25, 0.25)$, 30% have $V(j',:) = (0.25, 0.5, 0.25)$ and rest have $V(j'',:) = (0.25, 0.25, 0.5)$. We create the user feature matrix $U$ similarly having a $60 : 30 : 10$ split and the vectors spanning $U$ are only of the type that spans the simplex, i.e $U(i,:) = (1,0,0)$, $U(i',:) = (1,0,0)$ and $U(i'',:) = (1,0,0)$. Again, at every timestep $t$ the resulting matrix $M_t = UD_tV^\mathsf{T}$ is generated where $D_t$ is a randomly-generated diagonal matrix. So, $M_t$ is such that algorithms that quickly find the easily identifiable hott-topics perform very well. From Figure 2(b) we can see that $\mathtt{LRA} - \mathtt{EXP3}$, and $\mathtt{LRA} - \mathtt{UCB1}$ again outperform all the other algorithms. Their regret curve flattens much before all the other algorithms indicating that they have learned the best items for each user. The matrix completion algorithm $\mathtt{NMF} - \mathtt{Ban}$ again fails to get a reasonable approximation of $M$ and performs poorly. Also, we see that both the independent user model algorithms $\mathtt{RBA} - \mathtt{UCB1}$ and $\mathtt{RBA} - \mathtt{EXP3}$ perform poorly as the number of users and the number of items per user is too large and the independent base-bandits ($\mathtt{RBA}$) are not sharing information between themselves. In both the synthetic datasets, we see that stochastic column learning algorithm ($\mathtt{UCB1}$) is outperforming adversarial column learning algorithm ($\mathtt{Exp3}$) as the user preference over the best item is not changing over time. This has also been observed by Radlinski et al. (2008).

### 6.4. Real World Experiment 3

We conduct the third experiment to test the performance of LRA when our modelling assumptions are violated. We use the Jester dataset (Goldberg et al., 2001) which consist of over 4.1 million continuous ratings of 100 jokes from 73,421 users collected over 5 years. In this dataset there are many users who rated all jokes and we work with these users. Hence the user-item preference matrix is fully observed and we will not have to complete it using matrix completion techniques. Hence, this approach is very real world. We sample randomly 2000 users (who have rated all jokes) from this dataset and use singular value decomposition (SVD) to obtain a rank 4 approximation of this user-joke rating matrix $M$. In the resultant matrix $M$, most of the users belong to the four classes preferring jokes 99, 93, 96 and 28, while a very small percentage of users prefer some other jokes. Note, that this condition results from the fact that this real-life dataset does not have the hott-topics structure. The rank 4 approximation of $M$ of is shown in Figure 3(a), where we can clearly see the red stripes spanning the matrix indicating the low-rank structure of $M$. Furthermore, in this experiment we assume that the noise is independent

Bernoulli over the entries of $M$ and hence this experiment deviates from our modeling assumptions. From 3(b) again we see that $\mathtt{LRA} - \mathtt{EXP3}$, and $\mathtt{LRA} - \mathtt{UCB1}$ outperform other algorithms. Although the cumulative regret of $\mathtt{NMF} - \mathtt{Ban}$ is less than our proposed approaches, note that it does not converge and find the $d$ best items.

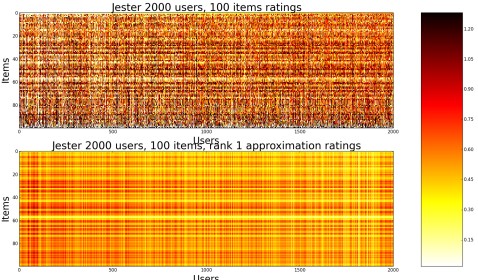

(a) Expt-3: 2000 Users, 100 items, Rank 1 approximation of Jester Dataset

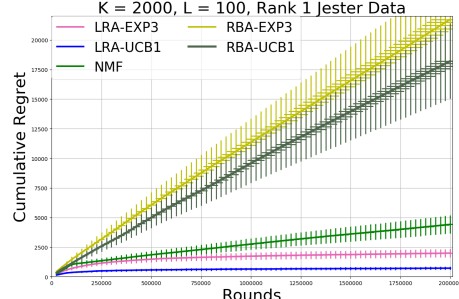

(b) Expt-3: Cumulative regret of different algorithms

*Figure 3.* A comparison of the cumulative regret in Jester Dataset

## 7. Related Work

Our work lies at the intersection of several existing areas of research, which we survey below.

**Bandits for Latent Mixtures:** The existing algorithms in latent bandit literature can be broadly classified into two groups: the online matrix completion algorithms and the independent user model algorithms. The *online matrix completion algorithms* try to reconstruct the user-item preference matrix $M$ from a noisy realization combining different approaches of online learning algorithms and matrix factorization algorithms. The NMF-Bandit algorithm in Sen et al. (2016) is an online matrix completion algorithm which is an $\epsilon$-greedy algorithm that tries to reconstruct the matrix $M$ through non-negative matrix factorization. Note, that this approach requires that all the matrices satisfy a weak statistical Restricted Isometry Property, which is not always feasible in real life applications. Another approach is that of Gopalan et al. (2016) where the authors come up with an algorithm which uses the Robust Tensor Power (RTP) method

of Anandkumar et al. (2014) to reconstruct the matrix $M$, and then use the OFUL procedure of Abbasi-Yadkori et al. (2011) to behave greedily over the reconstructed matrix. But the RTP is a costly operation because the learner needs to construct a matrix of order $L \times L$ and $L \times L \times L$ to calculate the second and third order tensors for the reconstruction. A more simpler setting has also been studied in Maillard & Mannor (2014) where all the users tend to come from only one class and hence this approach is also not quite realistic.

The second type of algorithms are the *independent user model algorithms* where for each user $i \in [K]$ a separate instance of a base-bandit algorithm is implemented to find the best item for the user. These base -bandits run independent of each other without sharing any information. These base-bandits can be any randomized algorithms suited for the adversarial setting or stochastic algorithms which tend to perform better under stochastic feedback assumptions.

**Ranked Bandits:** Bandits have been used to rank items for online recommendations where the goal is is to present a list of $d$ items out of $L$ that maximizes the satisfaction of the user. A popular approach is to model each of the $d$ rank positions as a Multi Armed Bandit (MAB) problem and use a base-bandit algorithm to solve it. This was first proposed in Radlinski et al. (2008) which showed that query abandonment by user can also be successfully used to learn rankings. Later works on ranking such as Slivkins et al. (2010) and Slivkins et al. (2013) uses additional assumptions to handle exponentially large number of items such that items and user models lie within a metric space and satisfy Lipschitz condition.

**Ranking in Click Models:** Several algorithms have been proposed to solve the ranking problem in specific click models. Popular click models that have been studied extensively are Document Click Model (DCM), Position Based Click Model (PBM) and Cascade Click Model (CBM). For a survey of existing click models a reader may look into Chuklin et al. (2015). While Katariya et al. (2017), Katariya et al. (2016) works in PBM, Zoghi et al. (2017) works in both PBM and CBM. Finally, Kveton et al. (2017) can be viewed as a generalization of rank-1 bandits of Katariya et al. (2016) to a higher rank. Note, that the theoretical guarantees of these algorithms does not hold beyond the specific click models.

**Online Sub-modular maximization:** Maximization of submodular functions has wide applications in machine learning, artificial intelligence and in recommender systems (Nemhauser et al., 1978), (Krause & Golovin, 2014). Intuitively, a submodular function states that after performing a set $A$ of actions, the marginal gain of another action $e$ does not increase the gain for performing other actions in $B \setminus A$. Online submodular function maximization has been studied in Streeter & Golovin (2009) where the authors propose a

general algorithm whereas Radlinski et al. (2008) can be considered as special case of it when the payoff is only between $\{0, 1\}$. Also, in the contextual feature based setup online submodular maximization has been studied by Yue & Guestrin (2011). An interesting property of submodular function is that a greedy algorithm using it is guaranteed to perform atleast $\left(1 - \frac{1}{e}\right)$ of the optimal algorithm and this factor $\left(1 - \frac{1}{e}\right)$ is not improvable by any polynomial time algorithm (Nemhauser et al., 1978). Note that the $\max$ function is a submodular function which satisfies the condition of monotonicity and submodularity.

## 8. Conclusions

In this paper, we studied the problem of suggesting a diverse list of items to users, with the best permutation of those items for individual users. The best permutation of items for an user contains its preference for the items in descending order with the best item at rank position 1. We formulated the above problem as a personalized ranking problem and proposed the latent ranker algorithm for this setting. We also evaluated our proposed algorithm on several simulated and real-life datasets and show that it outperforms the existing state-of-the-art algorithms.

There are several directions where this work can be extended. Instead of just providing an informal guarantee that the regret should be bounded, we intend to rigorously derive the regret of LRA (possibly a sub-linear regret). Note, that observing $d$ items at every timestep is helping LRA to learn more efficiently. Hence, while keeping the hott-topics assumption it is worthwhile to study the personalized ranking setting when only 1 item is allowed to be suggested at every timestep $t$. Another interesting direction is to look at structures where there are hott-topics assumption on user matrix as well as item matrix or maybe even at structures beyond hott-topics.

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
