# OpenReview forum: "Latent Ranked Bandits"
_ICML.cc/2019/Workshop/RL4RealLife — Submitted to RL4RealLife 2019_

### Official Review · AnonReviewer1 · 2019-05-24
**Interesting combination of bandits but not well suited for most real-world recommendation problems**

**Rating:** 2
**Confidence:** 3

**Review:**

This paper presents a method called Latent Ranker (LRA) that uses a bandit approach to learn a low-rank matrix in a recommendation context. The type of factorization that is done assumes that there are a small set of items (“hott topics”) that can express all of the items. The algorithm has a set of bandits across the columns (ColAlg) that select which base items to put in a list to get feedback from. Then the RowAlg bandit is run to create a permutation on those items. The algorithm observes the feedback on those items and updates both algorithms based on the reward. The paper then suggests how the regret bounds could be constructed for the algorithm, though it doesn’t provide any specific proof. In the experiment section, two synthetic datasets are used to evaluate LRA using different bandit methods, EXP3 and UCB, for the ColAlg and WMA for the RowAlg. It is compared against just doing an NMF and the RBA algorithm (also with EXP3 and UCB variants). The synthetic experiments set up situations with 2 or 3 items that fulfill the “hott topic” characteristics of the model and show that the LRA approach performs the best in terms of handling regret. Results are also shown on the Jester dataset once it has been filtered to all users.

Overall, this paper doesn’t seem very well suited for a workshop on “real world” reinforcement learning. It doesn’t evaluate the approach in conditions that are anything like a real-world recommendation problems, only on limited simulations and a very small, heavily filtered dataset. This especially matters because the modeling approach makes a very large assumption that feedback is observed on all items and that there are a certain number of items from which all other items can be expressed as a convex combination. Also, as a recommendation algorithm, it is odd that essentially the “candidate generation” aspect is unpersonalized and only the ranking of those candidates is personalized.

Positives:
- Interesting combination of two sets of bandits.
- Good overview of some related work.

Negatives:
- The “real-world” dataset used is one that has been filtered to only users that have rated every item, which is unrealistic in a real-world setting.
- The model assumes that it will observe the preferences for all recommended items, which also isn’t typically true in real-world recommender systems. Given the assumption of the model on the structure of the items, this makes the problem easy to solve, but isn’t very realistic for real applications.
- The paper claims that it learns diverse rankings, but no evaluation or specific analysis about the diversity properties of the algorithm or its outputs are provided in the paper.

Suggested Revisions:
- Section 1: Please explain why the assumed structure that there are a small number of optimal items is a reasonable one in a recommendation context.
- Section 2: Provide a reference for “hott topics” when you introduce the term.
- Section 4.1: Where you say you “leave the implementation of ColAlg and RowAlg to the users”, it would be good to provide information about what expectations the LRA algorithm places on those implementations.
- Section 6.2 & 6.3: Using only rank 2 or 3 problems for the synthetic experiments is not super convincing. Also, you don’t specify what “d” would be in these situations (the size of the list to get feedback on).
- Section 6.4: The section name “Real World Experiment 3” makes it sound like there are 3 real-world experiments when there is really just one. Perhaps you can change it to “Real-World Experiment” Or “Experiment 3: Jester Dataset” or something.
- Section 6.4: The sentence “Hence, this approach is very real world” is not true. Do you mean that because there is full data you can simulate feedback on anything?
- Section 6.4: Explain why you need to down-sample the data to 2000 users. Are there computational bottlenecks in the system?
- Section 6.4: The note about NMF-Ban having less regret is hard to tell from Figure 3 b. It looks like it has higher regret than the LRA methods.
- Section 6: Given that one of the motivations (mentioned in the introduction and conclusion) was to have a diverse list, it would be good to show metrics related to diversity in it.

---

### Official Review · AnonReviewer2 · 2019-05-24
**Interesting problem, an algorithm development, no analysis, okay numerical results, some confusions.**

**Rating:** 3
**Confidence:** 4

**Review:**



The paper studies the problem of learning personalized ranked lists of diverse item for multiple users. Assuming that the set of the most preferred items of all users is small (the user-item preference matrix is low rank), the problem is formed as an online learning problem. The authors propose a latent ranker algorithm (LRA) to solve the problem. The authors outline a sketch for potential regret analysis and conduct simulations using both synthetic and real traces to evaluate the performance of the proposed algorithm.

The performance of the proposed algorithm does not always outperform existing algorithms. For example, in Fig. 1, the regret of NMF (an existing matrix completion) algorithm have significant lower regret compare to the proposed algorithm over the entire simulation duration of 1 million steps. The authors argue (Sec. 6.4) that NMF does not converge and find the best d items. However, the reason that the current implementation of NMF does not converge is that it is set to explore for cd(K+L) rounds by choosing items uniform randomly where c is set to 10. We can easily adjust c and do periodic exploration to achieve good empirical performance. In fact, we can see from Fig. 1(b), even with this simple NMF implementation, it has significantly lower regret compared to the proposed algorithms until 1 million steps.

The authors outline a theoretical analysis sketch, but do not prove performance bound. Therefore, the theoretical contribution is somewhat limited.

The algorithm assumes that d is known. I am curious about the performance of the algorithm when d is unknown.

There seems to be a few typos/mistakes in the current description.

In Sec. 4, last paragraph between Sec. 4.1, “If the arm was not one of the previously suggested columns, its reward is …”. That seems to be incorrect and should be  “If the arm was one of the previously suggested columns, its reward is …”. Also, why should we update arm \hat{l_k} with reward 0?

In Sec. 6.2, the bottom of the first paragraph, the authors state that “NMF-Ban performs better than the independent user model algorithms but is outperformed by LRA-EXP3, and LRA-UCB1”. That is not correct. NMF-Ban (significantly) outperform LRA algorithms throughout the simulation duration.

In Sec. 6.4, it is stated that “although the cumulative regret of NMF-Ban is less than our proposed approaches, …” That is not consistent with Figure 3(b) unless the legend is incorrect.


Last, I also have a few confusions related to the proposed algorithm, possibly due to my lack of background knowledge on ranked bandits. The ColAlg is clear: it tries to find the most popular d items (with appropriate exploration). The RowAlg is less clear to me. The authors state that “Since LRA observes the individual rewards of all recommended items, we can compute the reward of any permutations of J_t in row i_t$.” How does that work?  It seems that we need to update simultaneously both the user preference and M_t(i_t, J_t(k)). It is also not clear how r_i(i,J) is observed.

---

### Decision · Program_Chairs · 2019-05-28

Reject